# Placement Recommendations for Single Kinect-Based Motion Capture System in Unilateral Dynamic Motion Analysis

**DOI:** 10.3390/healthcare9081076

**Published:** 2021-08-21

**Authors:** Laisi Cai, Dongwei Liu, Ye Ma

**Affiliations:** 1Research Academy of Grand Health, Faculty of Sports Sciences, Ningbo University, Ningbo 315211, China; cailaisi1995@163.com; 2School of Information Management and Artificial Intelligence, Zhejiang University of Finance and Economics, Hangzhou 310018, China; dongwei.liu@zufe.edu.cn; 3National Joint Engineering Research Centre of Rehabilitation Medicine Technology, Fujian University of Traditional Chinese Medicine, Fuzhou 350122, China; 4Key Laboratory of Orthopaedics & Traumatology of Traditional Chinese Medicine and Rehabilitation (Fujian University of TCM), Ministry of Education, Fuzhou 350122, China

**Keywords:** Kinect, depth sensor, placement, dynamic movement analysis

## Abstract

Low-cost, portable, and easy-to-use Kinect-based systems achieved great popularity in out-of-the-lab motion analysis. The placement of a Kinect sensor significantly influences the accuracy in measuring kinematic parameters for dynamics tasks. We conducted an experiment to investigate the impact of sensor placement on the accuracy of upper limb kinematics during a typical upper limb functional task, the drinking task. Using a 3D motion capture system as the golden standard, we tested twenty-one Kinect positions with three different distances and seven orientations. Upper limb joint angles, including shoulder flexion/extension, shoulder adduction/abduction, shoulder internal/external rotation, and elbow flexion/extension angles, are calculated via our developed Kinect kinematic model and the UWA kinematic model for both the Kinect-based system and the 3D motion capture system. We extracted the angles at the point of the target achieved (PTA). The mean-absolute-error (MEA) with the standard represents the Kinect-based system’s performance. We conducted a two-way repeated measure ANOVA to explore the impacts of distance and orientation on the MEAs for all upper limb angles. There is a significant main effect for orientation. The main effects for distance and the interaction effects do not reach statistical significance. The post hoc test using LSD test for orientation shows that the effect of orientation is joint-dependent and plane-dependent. For a complex task (e.g., drinking), which involves body occlusions, placing a Kinect sensor right in front of a subject is not a good choice. We suggest that place a Kinect sensor at the contralateral side of a subject with the orientation around $30^{\circ }$ to $45^{\circ }$ for upper limb functional tasks. For all kinds of dynamic tasks, we put forward the following recommendations for the placement of a Kinect sensor. First, set an optimal sensor position for capture, making sure that all investigated joints are visible during the whole task. Second, sensor placement should avoid body occlusion at the maximum extension. Third, if an optimal location cannot be achieved in an out-of-the-lab environment, researchers could put the Kinect sensor at an optimal orientation by trading off the factor of distance. Last, for those need to assess functions of both limbs, the users can relocate the sensor and re-evaluate the functions of the other side once they finish evaluating functions of one side of a subject.

## 1. Introduction

*Three-dimensional* (3D) motion analysis is a systematic study of human movement. With the quantification of joint kinematics, augmented by high-end instrumentation, investigators are able to obtain thorough 3D information on the movements of body segments through space and time, including linear and angular displacements, velocities, and accelerations. Three-dimensional motion analysis is widely used in studying the human neuromusculoskeletal system [1], assisting sports training [2], determining risk factors of musculoskeletal injury [3], diagnosing pathologies and planning treatment for individuals with musculoskeletal conditions [4], providing feedback for rehabilitation retraining [5], or assisting design of prosthetics or robotics [6].

*The 3D motion capture system* (3DMC) is widely used and is regarded as the “gold standard” for 3D motion analysis; however, existing 3DMC systems, active (e.g., NDI, Waterloo, Ontario) or passive (e.g., Vicon, Oxford Metrics Ltd., Oxford, UK), are limited for out-of-the-lab assessment due to the high price, the requirements for large space and professional technicians, and the extensive time for equipment setup and data post-processing [7].

Kinect sensors [8], originally developed for motion-sensing games by Microsoft, can deliver reliable depth images at around 30 Hz by combining a *depth sensor* with an *RGB-color camera* (RGB-D). Kinect SDK features real-time skeletal tracking of 3D locations for skeletal joints, with its RGB-D sensor and human pose estimation algorithm [9]. Such low-cost, portable Kinect sensors achieved great popularity in motion analysis [10,11,12,13].

A series of static motion assessments were conducted by using a single Kinect sensor, including balance and posture assessment [10,14], foot postural tracking [13], or foot postural index assessment [15]. Single Kinect-based systems were also used in gait analysis [7,11,16,17,18], gait rehabilitation training [19,20,21], trunk and lower limb kinematics during dynamic balance test [10], or squatting [22] and jumping tasks [23]. The systems were also used in the kinematic assessment or rehabilitation training of upper limb functional tasks for healthy people [12] and for patients with motor disorders [22,24,25,26,27]. The single Kinect-based systems generally have good reliability in both spatial and temporal parameters and have better capability in assessing temporal parameters compared with spatial parameters [7,11,12,16,17]. Compared with the static pose assessment, performing dynamic motion analysis using a single Kinect-based system is more challenging because of motion artifacts [28].

Using a multi-Kinect system is one of the solutions to improve the kinematic accuracy in dynamic motion analysis. Yang et al. [29] developed a multiple-Kinect system by combining results from multiple sensors with a nonlinear least square method. The system showed better performance in dealing with body occlusion and had better average accuracy of tracking human joints. Ryselis et al. [30] used algebraic operations in vector space in monitoring postures in physical training by using a three Kinect-sensor-based system. The multi-Kinect system demonstrated an increase of 15.7% in measuring kinematic parameters [30].

There are several vital limitations for those multi-sensor systems, which results in them being difficult to use in clinic, community, or home-based applications. First, the multi-Kinect systems need careful calibration before motion capture sessions. After the calibration, all Kinect sensors will have their own 3D coordinates in the laboratory coordination system. The Kinect sensors should not be moved once calibrated [29,30]. Second, the multi-Kinect systems also require robust and efficient software for data synchronization and fusion [29,30]. Third, the configuration and calibration of the systems require larger areas and a series of devices, including computers for each Kinect, servers for synchronization, calibration devices, and so on [29,30]. Those add-in requirements make the multi-Kinect-based systems semi-professional, less convenient, and not easy to use, limiting its wide use in out-of-the-lab assessment. To this end, a single-Kinect sensor-based motion analysis system is one of the appropriate solutions for out-of-the-lab motion analysis.

The placement of a Kinect sensor significantly influences the performance of measuring kinematic parameters, especially for dynamic tasks. A Kinect sensor is usually placed around 2 to 2.5 m in front of a subject, around one meter above the ground, with zero tilt angle in the horizontal plane to capture static postures [10,23,31]. Xu Xu et al. [32] found that placing the Kinect sensor in front of the participants was better than placing it $15^{\circ }$ or $30^{\circ }$ to one side during the computer-using task. Galna et al. [25] suggested that posing at a $45^{\circ }$ angle to the Kinect may improve the spatial accuracy for standing trunk flexion, hand clasping, and finger tapping tasks. The $45^{\circ }$ angle also had better accuracy in distinguishing a foot from the floor and determining knee location when the leg was straight [25]. Seo et al. [33] found that measuring the upper limb range of motion was most accurate when the Kinect was elevated $45^{\circ }$ in front of the subject, tilted toward the subject during the reaching task. For gait analysis, a Kinect sensor was mostly placed around one meter above the ground and tilted $0^{\circ }$ [17,31] to $-5^{\circ }$ [34] in the horizontal plane, with the sensor placed in front of the subject to provide a frontal plane view. Yeung et al. evaluated the accruacies of two versions of Kinect sensors for measuring kinematics during treadmill walking at five camera viewing angles (from $0^{\circ }$ to $90^{\circ }$) [35]. They found that at frontal viewing angle $0^{\circ }$, Kinect v2 sensor had better performance in capturing hip and knee saggital angle than other viewing angles [35]. To the best of our knowledge, there is little guideline or recommendation that thoroughly instructs a user how to place a Kinect sensor for dynamic tasks, especially for upper limbs using a single Kinect-based system.

We conducted a series of experiments to evaluate the impact of sensor placement on the accuracy of a single Kinect-based system in measuring upper limb kinematics during performing a typical upper limb functional task, the drinking (or hand to mouth) task. Drinking is one of the most common active daily activities. The ability to perform drinking and similar tasks represents movement coordination to some extent. The drinking task is a vital movement for assessing upper limb functions and evaluating the rehabilitation effects [36,37,38]. Compared with previous studies, the drinking task is more complex because it involves movements of multiple joints in multiple planes. More importantly, when performing the drinking task a subject’s upper arm or part of it is always occluded by the forearm. Using the drinking task as an example to evaluate the performance of the Kinect sensor can generalize to other upper limb functional tasks, which involve complex movements and body occlusions.

The purpose of our study is to investigate the sensor placement guideline for the single Kinect-based systems in measuring joint kinematics when performing upper limb functional tasks, represented by the drinking task. The guideline is investigated by evaluating the upper limb kinematic accuracy when the sensor is placed at twenty-one places with different distances and orientations. We also would like to give some general recommendations on the placement of Kinect sensors for motion analysis of all kinds of dynamic tasks.

## 2. Materials and Methods

### 2.1. Subject

We recruited ten healthy male college students (age: 25.3 ± 3.6 years old, height: 176 ± 4.2 cm, mass: 66.3 ± 4.5 Kg) who have no upper limb musculoskeletal disorders in six months prior to the experiment. The subjects all volunteered to participate in the study, and all signed informed consent forms before the experiment. The experiment protocol was approved by the Ethics Committee of Research Academy of Grand Health in Ningbo University.

### 2.2. Experiment Procedures

We conducted an experiment to compare the accuracy of upper limb joint kinematics using a single Kinect-based system when the sensor is placed at different locations. A state-of-the-art 3DMC system (Vicon, Oxford Metrics Ltd., Oxford, UK) is employed to generate the golden standard kinematic data. Each subject performed a series of hand-to-mouth task in the biomechanics laboratory of Ningbo University. An experimenter demonstrated the standard hand-to-mouth task for each subject to standardize the movement’s posture and speed. The hand-to-mouth (drinking) task represents activities such as eating and reaching the face. Subjects start with the arm in the anatomical position with their hand beside their body and end up with their hand reaching each their mouth (see Figure 1). Any shiny objects, such as watches, were removed from subjects to prevent interference with Kinect’s motion detection.

Twenty-one locations are evaluated for the single Kinect system. The locations differed by the distance between a sensor and a subject and by the Kinect’s orientations relative to the subject. Those two factors are easy to adjust in real-world environments. See Figure 2 for the details of Kinect placements. We denote the joint angles derived from a Kinect-based system located at different places by $K_{i,j}$ that at a distance of *i* meters and an orientation of *j* degrees. i=1.5,2.0,3.0, which represents the distance between a sensor and a subject is 1.5 m, 2.0 m, or 3 m. j=−60,−45,−30,0,30,45,60, in which −60 represents $60^{\circ }$ to a subject’s left side, −45 represents $45^{\circ }$ to a subject’s left side, −30 represents $30^{\circ }$ to a subject’s left side, 0 represents right in front of a subject, 30 represents $30^{\circ }$ to a subject’s right side, 45 represents $45^{\circ }$ to a subject’s right side, 60 represents $60^{\circ }$ to a subject’s right side. The Kinect sensor is placed on a tripod, 1 m above the ground.

We concurrently recorded the 3D coordinates of all reflective markers using the 3DMC and the coordinates of the skeletal joints using the Kinect v2 system. The coordinates of the reflective markers are recorded via the Nexus software at a sampling frequency of 100 Hz. The coordinates of the Kinect skeleton are recorded by a self-developed software based on Kinect SDK 2.0 at a sampling frequency of 30 Hz. Before the experiment, reflective markers were attached to the anatomical landmarks of each subject according to the UWA marker set [39]. Each subject firstly performed a static trial in the anatomical position. The elbow and wrist markers were then removed during the dynamic trials. Each subject sat in a chair and performed the hand-to-mouth task at least five times for each Kinect position. Subjects were provided with rest breaks between Kinect relocation.

### 2.3. Data Analysis

Upper limb joint angles, including shoulder flexion/extension, shoulder abduction/adduction, shoulder internal/external rotation, and elbow flexion/extension are calculated for the 3DMC system and the Kinect-based system, respectively.

For the Kinect-based system, Upper limb joint angles are calculated based on the 3D coordinates of trunk and upper limb joint center (see Figure 3, left), including ShoulderRight, ShoulderLeft, SpineShoulder, SpineMid, ElbowRight, and WristRight derived from Kinect v2 SDK. The 3D coordinates are pre-processed by a zero-lag fourth-order Butterworth low-pass filter with a cut-off frequency of 6 Hz. According to the recommendation of R. Bartlett, 4 to 8 Hz are often used as the cut-off frequencies in low-pass filtering movement data [40]. We carried out a series of residual analyses on the raw data using the cut-off frequencies of 4, 5, 6, 7, and 8 Hz, respectively. We selected 6 Hz as the cut-off frequency because it yields the best result in our task and is validated for upper limb function assessment [12].

*Local segment coordination* (LSC) systems of the torso and upper arm (taking the right arm as an example) are defined in Table 1. The kinematic model of Kinect Φ calculates three Euler angles for shoulder joint following the flexion (+)/extension (−) adduction (+)/abduction (−) and internal (+)/external (−) rotation order. The shoulder angles are denoted as $\alpha_{\mathrm{FE}}$, $\alpha_{\mathrm{AA}}$, and $\alpha_{\mathrm{IE}}$. The elbow flexion angle, denoted as $\alpha_{\mathrm{EFE}}$, is calculated using the trigonometric function by the position coordinates from ShoulderRight, ElbowRight, and WristRight. The model Φ is developed using Matlab 2018a.

For the 3DMC system, joint angles are calculated via the UWA kinematic model (denoted as Γ) at the capture and analysis platform, Nexus. The UWA model is coded using *body language* [39]. The UWA marker set includes 18 markers (see Figure 3, right). Trunk, upper arm, and forearm segments are defined based on these markers. The shoulder joint center was determined by the posterior shoulder marker (PSH) and the anterior shoulder marker (ASH). The elbow joint center is determined by the medial elbow epicondyle (EM) and lateral elbow epicondyle (EL) markers. The wrist joint center is determined by the ulnar styloid (US) and radial styloid (RS) markers. We use the calibrated anatomical systems technique [39] to establish motions of anatomical landmarks regarding to the coordination systems of the upper-arm cluster (PUA) or the forearm cluster (DUA). Thus, the motion of the upper-limb landmarks could be reconstructed from their constant relative positions in the upper-arm technical coordinate system. Shoulder joint angles, including flexion (+)/extension (−) $\beta_{\mathrm{FE}}$, adduction (+)/abduction (−) $\beta_{\mathrm{AA}}$, and internal (+)/external (−) rotation $\beta_{\mathrm{IE}}$, as well as elbow flexion (+)/extension (−) angle $\beta_{\mathrm{EFE}}$ are served as the golden standard.

### 2.4. Statistical Analysis

We extracted *angle at the point of target achieved* (PTA) values from the shoulder and elbow angular waveforms for both the Kinect and 3DMC system, which are denoted as $K_{\Phi }$ and $K_{\Gamma }$. The *mean absolute error* (MEA) $E_{\Phi ,\Gamma }=\left|K_{\Phi }-K_{\Gamma }\right|$ represents the accuracy of the Kinect-based system.

The Shapro–Wilk test was performed to test the normality of the data $E_{\Phi ,\Gamma }$ for all upper limb angles when the Kinect sensor was placed at 21 positions. A two-way repeated measure ANOVA was conducted to explore the impacts of distance (1.5 m, 2 m, and 3 m) and orientation (see the details of the seven orientations in Section 2.3) on the MEAs. A statistically significant difference is accepted as p<0.5. The eta squared ($\eta^{2}$) is used as the measure of effect size. The $\eta^{2}$ of 0.01, 0.06, and 0.14 means small effect, moderate effect, and large effect [41]. For those factors that reach significant difference, post hoc analysis is applied to show where the difference occurs.

## 3. Results

In Table 2, we presented descriptive statistics of the *mean absolute error* (MEA) of upper limb joint angles between the Kinect-based system and the golden standard system when the Kinect sensor is placed at different distances and orientations with respect to the subject. The result of two-way repeated measure ANOVA is also demonstrated in Table 2, which explores the impact of distance and orientation on the accuracy of upper limb kinematic measurement by the Kinect-based system.

The results of ANOVA for all four upper limb angles including shoulder *flexion/extension* (Flex/Ext), shoulder *abduction/adduction* (Abd/Add), shoulder *internal/external rotation* (IR/ER), and elbow flexion/extension angles are similar (see Table 2). There is a significant main effect for orientation (p<0.05,$\eta^{2}=0.21∼0.54$). The main effects for distance ($p=0.55∼0.99,\eta^{2}=0.06∼0.10$) and the interaction effects (p=0.34∼0.92,$\eta^{2}=0.12∼0.18$) do not reach statistical significance.

As the main effect for orientation reached significant difference for all four upper limb angles, post hoc comparisons using the LSD test were applied to investigate the performance of Kinect when placed at seven orientations. The results are visualized in Figure 4. It is clear that the performance in kinematic measurement is quite different when the Kinect is placed at different orientations. The impact of orientation differs for the four joint angles we investigated.

In terms of the shoulder flexion/extension angle (see Figure 4, upper left and Table 2), the angle at the orientation $0^{\circ }$, denoted by $K_{0}$, is significantly different in comparison with the other six orientations, including $30^{\circ }$, $45^{\circ }$, and $60^{\circ }$ to the left and right side of the subject, which are denoted by $K_{-30}$, $K_{-45}$, $K_{-60}$, $K_{30}$, $K_{45}$, and $K_{60}$ respectively. The Kinect being placed at $0^{\circ }$ has the largest error than being placed at other orientations with more than $10^{\circ }$ more deviations compared with the standards. There are no significant differences between MAEs of the Kinect at the one side of the subject, either left (p=0.53∼0.99) or right (p=0.46∼0.99).

The sagittal plane angle of the elbow joint, the elbow flexion/extension angle (see Figure 4, bottom right and Table 2), shows different results with the shoulder joint. The Kinect placed at $30^{\circ }$ and $45^{\circ }$ to the left ($K_{-30}$) show significantly smaller MEAs than $60^{\circ }$ to the left ($K_{-60}$). The MEAs of $45^{\circ }$ and $60^{\circ }$ to the right sides ($K_{45}$ and $K_{60}$) are significantly smaller than $60^{\circ }$ to the left ($K_{-60}$). The MEAs of $30^{\circ }$ to the right side ($K_{30}$) are significantly smaller than $60^{\circ }$ to the right side ($K_{60}$). There are no significant differences between the MAEs of other orientations (p=0.10∼0.92).The MEAs are smaller when the Kinect is placed slightly deviated from the position in front of the subject for both left and right side ($K_{-30}$ and $K_{30}$). The Kinect shows smaller MEAs when placed at the left side than the right side as well as in front of the subject ($K_{-60,-45,-30}$ < $K_{0,30,45,60}$).

In terms of the shoulder angle in the frontal plane, shoulder abduction/adduction angle (see Figure 4, upper right and Table 2), the MEA pattern is quite different compared to angles in the sagittal plane. The MEA is the smallest ($2.85^{\circ }$) when the Kinect is placed in front of the subject. The MEA is the largest at $60^{\circ }$ to the right side and shows significant difference with $30^{\circ }$ to the left (p=0.01), in front of the subject (p<0.01), and $45^{\circ }$ to the right (p=0.03). The MEAs of the left side ($K_{-60,-45,-30}=4.80∼6.74$) are smaller than the right side ($K_{0,30,45,60}=6.05∼9.38$).

For shoulder angle in the coronal plane, the shoulder internal/external rotation angle (see Figure 4, bottom left and Table 2), the MEAs are the largest at $60^{\circ }$ to the right side and show significant difference with other orientations (p=0.00∼0.02). The MEAs of the right side ($K_{0,30,45,60}=23.73∼36.83$) are significantly larger than the left side ($K_{-60,-45,-30}=8.29∼11.26$).

## 4. Discussion

We investigated the accuracy of a single Kinect-based system in measuring upper limb kinematics during a typical upper limb functional task when the Kinect sensor is placed at different distances and orientations. The upper limb joint angle, including shoulder flexion/extension, shoulder abduction/adduction, shoulder internal/external rotation, and elbow flexion/extension angle are simultaneously measured by a Kinect v2 sensor and a standard 3D motion capture system (3DMC). We evaluated the performance of a Kinect sensor being placed at twenty-one places via the mean absolute error (MEA) of each angle between a Kinect and a 3DMC system. We want to discover a guideline of Kinect placement for the Kinect users in assessing upper limb functional tasks. We also would like to summarize and provide some general recommendations on placement of a single Kinect sensor for all dynamic functional tasks.

### 4.1. Effects of the Kinect Placement

Our study finds that in the the sensor’s effective capture space, the distance between a subject and a sensor does not influence the kinematic accuracy for both shoulder joints and elbow joint in all degrees of freedom. The MEAs between the Kinect sensor and the golden standard have no significant differences among the distances of 1.5 m, 2 m, and 3 m for the drinking task. Dutta [3] investigated the feasibility of Kinect as a 3D motion capture system in the workspace by comparing the location accuracy of a simplified four-cubes system with a Vicon system. In Dutta’s study, the root-mean-squared errors between the two systems are around 0.0065 m, 0.0109 m, and 0.0057 m in the direction of *x*, *y*, and *z* axes (to the right, away from the sensor, and upward), respectively. The accuracy is at the same level for most places over the range of 1.0 m to 3.6 m, with the worse accuracy at the margin of the effective field of view in all three planes [3]. Although upper limb functional tasks are more complex than Dutta’s simplified system, both our study and Dutta’s showed similar results, indicating that besides close to the margin of the Kinect’s effective field of view, the distance between a sensor and a subject is not a sensitive factor to the accuracy of upper limb kinematics using a single Kinect-based system.

The Kinect shows smaller MEAs when placed at the investigated limbs’ contralateral side for our upper limb functional task. In our study, the subjects performed a right-side drinking task, which means that the the subject’s left side is the contralateral side. Our results (see Table 2, Figure 4) reveal that besides the shoulder flexion/extension angle, placing the Kinect at the left side of the subject showed lower kinematic errors than the right side; however, in a computer use task, which involves no upper limb occlusion, Xu Xu et al., found that placing the Kinect sensor in front of the subject is more accurate on measuring shoulder kinematics in comparing with placing the sensor $15^{\circ }$ or $30^{\circ }$ to the left [32]. In another upper limb joint angle measuring task using Kinect sensor, the subjects were asked to lift their right arm, and point their index finger to the targets [33]. For such a reaching task with little elbow flexion/extension and little occlusion between the upper limb and forearm, the Kinect sensor’s optimal place is quite different from our study. The Kinect sensor in front of the subject, with $45^{\circ }$ elevation showed the least errors in measuring upper limb range of motions [33].

The Kinect sensor is usually placed in front of the subject with zero or less than $5^{\circ }$ tilt in the horizontal plane [17,31,34] gait analysis and static posture assessments. Compared with gait analysis and static posture assessments, kinematic assessment for the upper limbs using Kinect sensors is far more challenging [42]. Upper limb functional activities show larger variations of execution in the healthy population (as opposed to the stereotyped gait pattern) related to less stringent task accomplishment and the higher degrees of freedom in the upper limb [42]. The upper limbs, especially the shoulder joint, have a very large working range, in comparison with the lower limbs. Furthermore, the upper limb joints can easily occlude each other. The placement of the Kinect sensor is, therefore, easier for gait analysis and static posture assessments.

There is not a single optimal measurement position for a Kinect sensor in measuring human postures. The optimal Kinect placement is task-dependent. The placement of the Kinect sensor should be investigated carefully for each functional task, especially for those with body occlusions. To those tasks with little body occlusions, such as gait analysis, static movement assessment, or upper limb tasks with little body occlusion, the Kinect sensor could be placed in front of the subject, with small range of tilt angle, or with some degrees of elevation, depending on the tasks. For those tasks involving body occlusions, the placement of the Kinect sensor is more challenging and vital. Researchers must carefully evaluate the sensor placement to ensure the system’s optimal performance.

The original Kinect uses *structured-light* (SL) technology (denoted as ${\mathrm{Kinect}}^{\mathrm{SL}}$) for obtaining depth information. The ${\mathrm{Kinect}}^{\mathrm{SL}}$ uses a low number of patterns to obtain depth estimation of the scenery at a relatively high sampling frequency (around 30 frames per second) [28]. The ${\mathrm{Kinect}}^{\mathrm{SL}}$ is based on the standard structured light principle and uses simple triangulation techniques to compute the depth information between the projected pattern seen by the *near infra-red* (NIR) laser camera and the input pattern stored on the unit.

The updated Kinect sensors are based on the *time-of-flight* principle (denoted as ${\mathrm{Kinect}}^{\mathrm{ToF}}$) [28,43]. ${\mathrm{Kinect}}^{\mathrm{ToF}}$ also is more robust and accurate in tracking human pose [44]. ${\mathrm{Kinect}}^{\mathrm{ToF}}$ would be a better choice for depth-stereo applications [28]. The ${\mathrm{Kinect}}^{\mathrm{ToF}}$ is based on measuring the time and light emitted by an illumination unit using the *continuous wave* (CW) intensity modulation approach [45]. The ToF-camera has a unique ambiguous measurement range due to the periodicity of the signal it uses. The human skeleton of the ${\mathrm{Kinect}}^{\mathrm{ToF}}$ is more anthropometric with smaller offsets of the skeleton joints, better overall accuracy of joint positions, more reliable even with partial body occlusions and so forth in comparison with the ${\mathrm{Kinect}}^{\mathrm{SL}}$ [44].

The ${\mathrm{Kinect}}^{\mathrm{ToF}}$ suffers from various artifacts caused by light reflection [45]. Besides the systematic errors of the range sensors, the sensors also suffer from errors from the “multi-path effect” and dynamic scenery [45]. The average depth accuracy is under 2 mm in the central viewing cone and increases to 2–4 mm in the range of up to 3.5 mm. The maximal range captured by ${\mathrm{Kinect}}^{\mathrm{ToF}}$ is around 4.0 m, where the average error increases beyond 4 mm [44]. In the effective field of view of a Kinect sensor, the Kinect sensor should be placed as near as possible. Although in our study, the upper limb kinematics are not significantly influenced by the “distance” factor.

### 4.2. Recommendations of the Kinect Placement for Dynamics Tasks

We propose the following recommendations for placing Kinect sensors in assessing kinematics of dynamic tasks. Researchers should firstly set an optimal effective capture space, which enables the maximal volume of view and make sure all investigated joints can be robustly captured in this space by setting the height and tilt angle of the sensor as well as the sensor orientation of the sensor relative to the investigated subject. The Kinect sensors use a depth sensor and a color camera, in which these two cameras have different fields of view; therefore, the effective capture space should consider both the depth sensor and the color camera.

For example, the depth sensor on the Kinect v2 has a horizontal *field of view* (FoV) of $70.6^{\circ }$ and a vertical FoV of $60^{\circ }$. If we place the Kinect v2 sensor at the height of 1.0 m and the tilt angle of $0^{\circ }$, the effective 3D capture space of the Kinect sensor could be demonstrated in Figure 5. The intersection line between the Kinect’s depth FoV and the ground is around 1.73 m from the sensor. The width of the line is around 2.46 m. Similarly, the horizontal and vertical FoVs of the color camera are around $84^{\circ }$ and $54^{\circ }$, respectively. The intersection line of the camera is around 1.97 m far from the Kinect sensor and the line width is around 3.56 m. In this scenario, if the researcher would like to assess dynamic motions, including foot motions, such as walking, run, or squat, the starting line of the capture space should be more than 1.73 m far from the sensor. If it is smaller than 1.73 m, the depth sensor of Kinect will miss the information of the foot joint and other lower limb joints depending on the distance. By visualizing the FoV space of the Kinect location setting, people can obtain all aforementioned information by trigonometric function calculations. For tasks such as walking, the most effective capture space could be along the the Kinect sensor’s face angle, with the distance to the Kinect sensor over the range of 1.73 m and 4 m. For tasks that only involve upper limbs, the effective capture space is larger with the distance between a subject and a Kinect sensor in the range of 1.5 m and 4 m, depending on the tasks investigated.

Secondly, the placement of the Kinect should ensure minimal body occlusion during the entire movement. For motions such as static motion assessment, gait analysis, or dynamic balance test, there is little body occlusion during whole tasks. The users only need to ensure that the subjects during the investigated movements are in the effective capture space; however, for upper limb functional tasks such as drinking, body occlusions occur when the subject flexes his/her elbow joint or his/her hand reaches his/her mouth. The depth information and the color images of upper arm are deviated because of the occlusion, which causes inaccuracy of the kinematic measures.

Thirdly, the out-of-the-lab environment is more challenging than the traditional laboratory for a 3D motion capture system due to the limited space, complex layout, or application scenario. In such a challenging task, the optimal location of the Kinect sensor sometimes cannot be achieved. The Kinect location should be compromised by trading off between several key factors such as distance and orientation. The orientations in our research are relative to the sagittal plane of the subject, in which the most common orientation, i.e., right in front of and facing toward the subject, is defined as $0^{\circ }$ and to the right side and to the left side of the subjects are defined as positive and negative orientations (see Figure 2). From our study, we found that in the effective FoV, the “distance” factor of the Kinect location has less influence on the kinematic measurements than the “orientation” factor; therefore, in a challenging scenario, the most important Kinect location is one that ensures the optimal orientation.

Lastly, for those functional tasks involving bi-lateral limbs, we cannot find an optimal location where the system has good accuracy for both left and right sides using a single depth-sensor-based system. If the users need to assess the kinematics of both sides, they can relocate the sensor and re-evaluate the functions of the other side once they finish evaluating the kinematic functions of one side of a subject.

## 5. Conclusions

The placement of a Kinect sensor is of great importance for assessing joint kinematics when performing dynamic tasks using a single Kinect-based system. We found that placing a Kinect sensor right in front of a subject is not a good choice for complex upper limb tasks (such as the drinking task), which involves body occlusions. We suggest placing a Kinect sensor at the contralateral side of a subject with the orientation around $30^{\circ }$ to $45^{\circ }$ for upper limb functional tasks. For all kinds of dynamic tasks, we put forward the following recommendations on the placement of a Kinect sensor. First, set an optimal location for capture, making sure that all investigated joints are visible during the whole task. Second, sensor placement should avoid body occlusion at the maximum extension by setting the height and tilt angle of the sensor as well as the sensor orientation of the sensor relative to an investigated subject. Third, if an optimal location cannot be achieved in an out-of-the-lab environment, researchers could put the Kinect sensor at an optimal orientation by trading off the factor of distance. Last, for those who need to assess functions of both limbs, the users can relocate the sensor and re-evaluate the functions of the other side once they finish evaluating the functions of one side of a subject.

## Figures and Tables

**Figure 1 healthcare-09-01076-f001:**
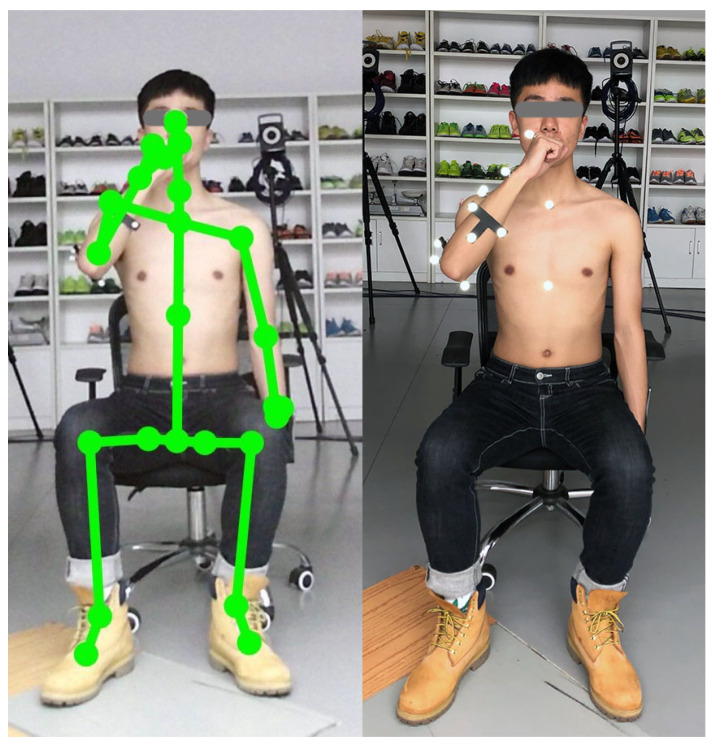
A subject performing a hand-to-mouth task. (**Left**) The subject with the Kinect Skeleton derived from Kinect SDK. (**Right**) The subject with reflective markers attached to anatomical landmarks.

**Figure 2 healthcare-09-01076-f002:**
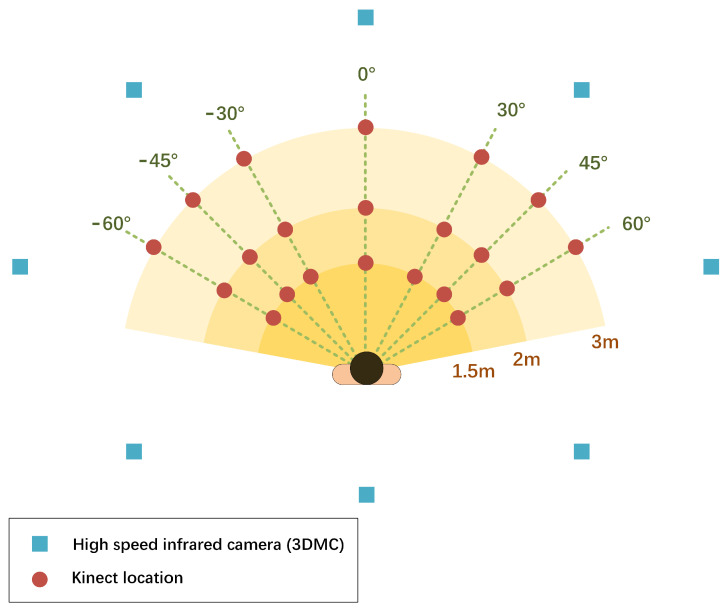
Kinect locations specified by distance and orientations relative to the subject. A Kinect sensor is placed at 21 locations at three different distances and seven orientations. Specifically, a Kinect sensor is place at 1.5 m, 2 m, and 3 m far from a subject. For each distance, a Kinect sensor is placed at seven orientations, right in front of the subject $0^{\circ }$, $60^{\circ }$, $45^{\circ }$, and $30^{\circ }$ to the left side (represented by negative values) and right side of a subject (represented by positive values).

**Figure 3 healthcare-09-01076-f003:**
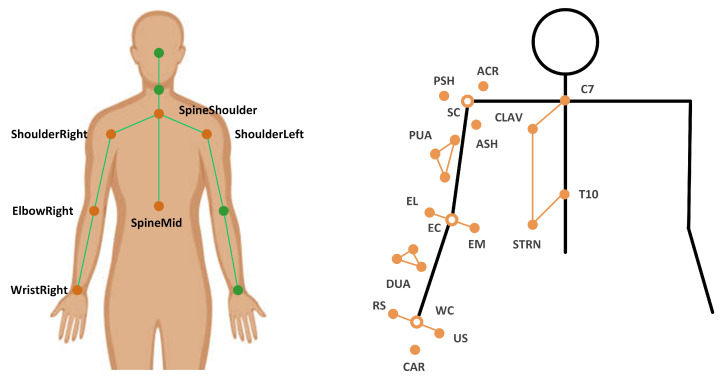
Kinect skeleton joints (**left**) and UWA marker set (**right**).

**Figure 4 healthcare-09-01076-f004:**
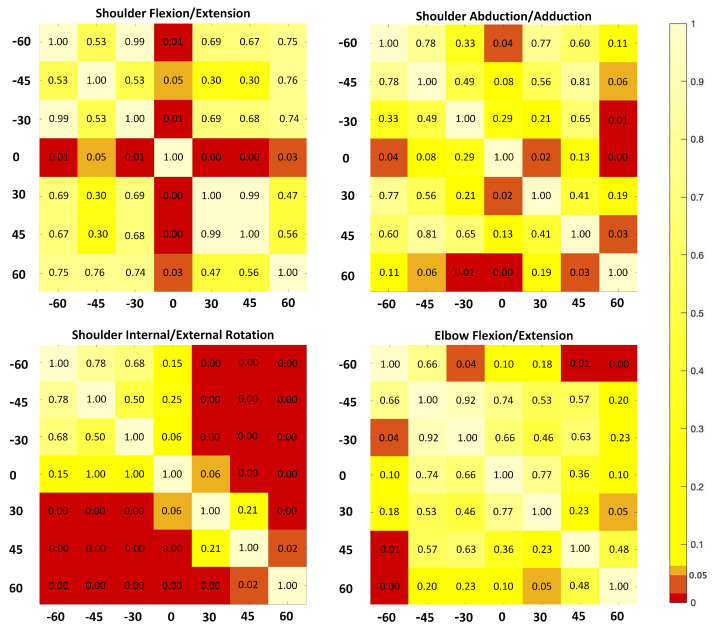
Post hoc statistical analysis on mean absolute errors (MAEs) of upper limb kinematics derived from a single Kinect-based system when the sensors are placed at seven orientations. The significant levels of MEAs between any two orientations are visualized. The colors from orange to red mean that the significant levels are lower than 0.05.

**Figure 5 healthcare-09-01076-f005:**
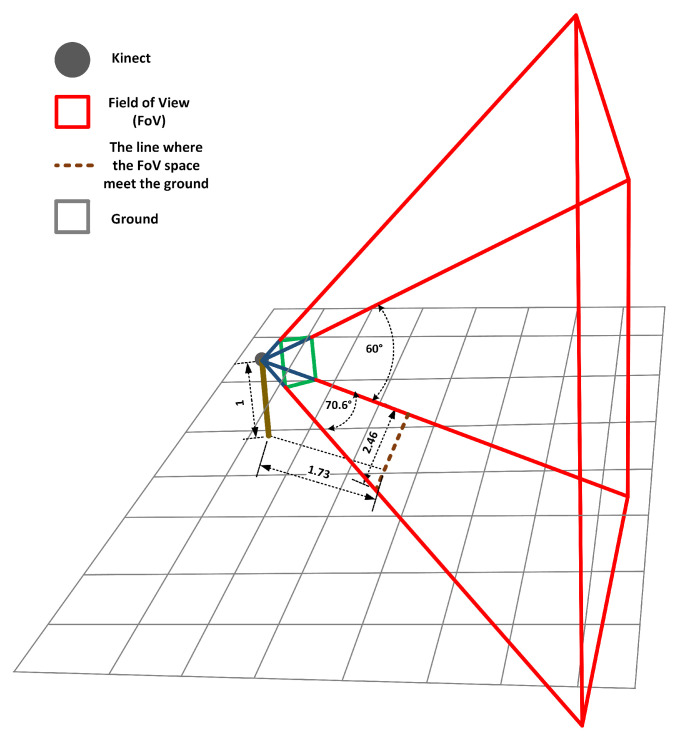
Demonstration of the effective capture space of a Kinect depth sensor. In this scenario, the Kinect sensor (represented by a black circle) is placed at a tripod 1 m above the groud (represented by grey grids) with $0^{\circ }$ title. The demo uses the depth sensor of Kinect v2 as the example. The horizontal and vertical FoV of the depth sensor are $70.6^{\circ }$ and $60^{\circ }$, respectively. The intersection line of the FoV space and the ground is around 1.73 m far from the sensor and the width of the intersection line is around 2.46 m.

**Table 1 healthcare-09-01076-t001:** Definition of the upper arm and torso anatomical segment coordinate system for the Kinect system.

Name		Definition
Torso	Origin	SpineShoulder
	X	Unit vector perpendicular to two vectors (Y and the vector from ShoulderRight to ShoulderLeft)
	Y	Unit vector going from SpineMiddle to SpineShoulder
	Z	Unit vector defined by the X-axis and the Y-axis to create a right-hand coordinate system
Upper Arm	Origin	The elbow joint center (ElbowRight)
	X	Unit vector perpendicular to the Y-axis and the Z-axis, pointing anteriorly
	Y	Unit vector going from SpineMiddle to SpineShoulder
	Z	Unit vector defined by the X-axis and the Y-axis to create a right-hand coordinate system

**Table 2 healthcare-09-01076-t002:** Descriptive statistics of the mean absolute error of upper limb joint angles via the Kinect-based system in comparison with the Golden standard system and results of the two-way ANOVA in effects of location, orientation and their interaction (Distance * Orientation) on mean absolute error.

Joint Angle	Distance	Mean (SD)	Orientation	Mean (SD)	*p* Value of ANOVA		
					Main Effect	*p* Value	eta
Shoulder Flex/Ext	1.5	27.96 (5.02)	−60	26.17 (5.93)	Distance	0.94	0.10
			−45	29.63 (11.06)			
	2	27.12 (4.41)	−30	24.79 (10.87)	Orientation	0.05	0.21
			0	36.66 (10.61)			
	3	25.04 (6.55)	30	23.71 (8.82)	Distance * Orientation	0.78	0.12
			45	23.02 (9.98)			
			60	23.34 (7.56)			
Shoulder Abd/Add	1.5	6.11 (2.43)	−60	4.80 (3.05)	Distance	0.85	0.09
			−45	6.56 (5.29)			
	2	6.26 (3.14)	−30	6.74 (4.19)	Orientation	0.02	0.24
			0	2.85 (1.71)			
	3	6.41 (1.95)	30	7.33 (7.46)	Distance * Orientation	0.92	0.14
			45	6.05 (3.56)			
			60	9.38 (8.49)			
Shoulder IR/ER	1.5	18.93 (10.56)	−60	8.29 (7.29)	Distance	0.99	0.06
			−45	11.26 (11.18)			
	2	19.70 (11.32)	−30	9.42 (8.15)	Orientation	0.02	0.22
			0	16.91 (9.26)			
	3	19.27 (11.92)	30	23.73 (10.8)	Distance * Orientation	0.85	0.15
			45	28.86 (10.84)			
			60	36.83 (15.17)			
Elbow Flex/Ext	1.5	22.49 (7.65)	−60	21.46 (8.73)	Distance	0.55	0.08
			−45	22.61 (11.14)			
	2	21.61 (9.20)	−30	10.20 (5.75)	Orientation	0.05	0.54
			0	21.37 (6.46)			
	3	19.27 (11.92)	30	18.91 (7.66)	Distance * Orientation	0.34	0.18
			45	25.54 (7.43)			
			60	28.32 (4.83)			

## Data Availability

The data are available from the corresponding author upon request.

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
