# Peer review of "Placement Recommendations for Single Kinect-Based Motion Capture System in Unilateral Dynamic Motion Analysis"

_healthcare, 2021, doi:10.3390/healthcare9081076_

Round 1

Reviewer 1 Report

Line 173: For the cut-off frequency, residual analysis is used. You should elaborate on that.

Your title of the paper is:

Placement recommendations for single Kinect based motion capture system in dynamic motion analysis 

This means that your recommendations are valid for any 'dynamic motion analysis'. However, this is not true. When one has a simultaneous movement of the upper limbs, he needs to simultaneously analyze both arms’ joints. Your suggestion mentioned in lines 24 to 26, 394-398, and 412 to 414 is not valid for simultaneous upper limbs movements.

Author Response

Line 173: For the cut-off frequency, residual analysis is used. You should elaborate on that.

Response: thank you for this recommendation. We have elaborated the “cut-off frequency” issue in our revised manuscript (see Line 172-177). We added the following sentences “According to the recommendation of R. Bartlett, 4 to 8 Hz are often used as the cut-off frequencies in low-pass filtering movement data [40]. We did a series of residual analyses on the raw data using the cut-off frequencies of 4, 5, 6, 7, and 8 Hz, respectively. We selected 6 Hz as the cut-off frequency because it yields the best result in our task and is validated for upper limb function assessment [12].”

Your title of the paper is:

Placement recommendations for single Kinect based motion capture system in dynamic motion analysis 

This means that your recommendations are valid for any 'dynamic motion analysis'. However, this is not true. When one has a simultaneous movement of the upper limbs, he needs to simultaneously analyze both arms’ joints. Your suggestion mentioned in lines 24 to 26, 394-398, and 412 to 414 is not valid for simultaneous upper limbs movements.

Response: we have revised our title to “Placement recommendations for single Kinect based motion capture system in unilateral dynamic motion analysis.

Reviewer 2 Report

The scientific contribution and novelty of the paper is not clearly evidenced. The authors just included some phrases that not improve the novelty and scientific soundness of the article.

Author Response

The scientific contribution and novelty of the paper is not clearly evidenced. The authors just included some phrases that not improve the novelty and scientific soundness of the article.

Response: thank you for your time. We believe that we clarified our contribution in the last response letter. “Based on our literature review (including both Xu Xu et al.’s study and Galna et al.’s study), there is no study thoroughly investigated the effect of “orientation” and “distance” on the kinematic accuracy for a single Kinect based system during assessing dynamic tasks, especially for complex tasks and movements with body occlusions. There is also no guideline for placement of a depth sensor for the single-Kinect based systems in joint kinematic assessment during upper limb functional tasks. There are also no depth sensor placement guideline for all kinds of dynamic tasks. By our study, we gave a guideline for Kinect placement in assessment of upper limb functional tasks. We also summarized and gave general recommendations on placement of a single-Kinect sensor for all dynamic functional tasks.” We believe that users of single-Kinect based system will benefit from our study.

Round 2

Reviewer 1 Report

The authors have modified the manuscript according to my feedback. 

This manuscript is a resubmission of an earlier submission. The following is a list of the peer review reports and author responses from that submission.

Round 1

Reviewer 1 Report

The authors have recruited ten male college students to study the effect of the placement of Kinect V2 sensor on the measurement accuracy of dynamic tasks. Unilateral hand to mouth drinking task which involves upper limb movements such as shoulder flexion/extension are used in the protocol. Based on the results, the authors suggest general recommendations on the placement of the Kinect V2 sensor for the motion analysis of dynamic tasks.

I think that the approach for doing this study is interesting. However, I have some comments which I would include in the following:

  1. The introduction is not including most of the previous studies for review.
  2. Line 109, line 257, and ...: It is stated that the purpose of the study is to give a general guideline for the placement of the Kinect sensor. The authors should remember that the movements of the body parts such as arms and feet is not always unilateral. In many situations, we require bi-lateral movements the arms, for example. Then, your recommendations is not valid anymore.
  3. Line 118: The participants should also include unhealthy individuals such as those with paralysis. In the case of such participants, the movement is not as smooth as the healthy individuals which affects the dynamic movement and consequently the measurement accuracy.
  4. Line 143 and Figure 5: The Kinect is placed on a tripod one meter above the ground. In previous publications, it is shown that the position of Kinect at higher locations than the moving part of the body results in better accuracies. Why did not you consider this possibility?
  5. Line 163: You have set the cut-off frequency to 6 Hz. Is this frequency high enough to cover the velocity range of the movement?
  6. Table 1: The results for repeated movements are included in the table. How could you guarantee the exact movement of the repeated tasks at different distances and angles?

Reviewer 2 Report

The paper tries to answer a question that is solved. Although the introduction is quite complete and includes a large amount of evidence in this regard, they themselves indicate the answer they were looking for in their work by mentioning the article by Galna et al [22] and Xu Xu et al. [28] those who responded to a similar problem using a single Kinect sensor 7 and 4 years ago respectively, with similar results and conclusions.

A easy way to solve that it is include an additional kinect (low cost solution).

On the other hand, the Kinect sensors, although they are still available on the market, are a technology that has not been produced by Microsoft for quite some time, which makes it difficult to incorporate them into rehabilitation centers or medical centers. Remember that there are a large number of low-cost and better-performing options to perform motion analysis of repetitive tasks with and without risk of occlusion.
Regarding the methodology and results of the work, they are correct, however, the scientific contribution and novelty of the paper is not clearly evidenced.